# How well are non-communicable disease services being integrated into primary health care in Africa: A review of progress against World Health Organization's African regional targets

**Azeb Gebresilassie Tesema**[1]*, **Whenayon Simeon Ajisegiri**[1], **Seye Abimbola**[1,2], **Christine Balane**[1], **Andre Pascal Kengne**[3], **Fassil Shiferaw**[4], **Jean-Marie Dangou**[5], **Padmanesan Narasimhan**[6], **Rohina Joshi**[1,2,7‡], **David Peiris**[1‡]

1 The George Institute for Global Health, University of New South Wales (UNSW), Sydney, Australia, 2 School of Public Health, University of Sydney, Sydney, Australia, 3 Non-communicable Diseases Research Unit, South African Medical Research Council & University of Cape Town, Cape Town, South Africa, 4 World Health Organization, Ethiopia Office, Addis Ababa, Ethiopia, 5 World Health Organization, Regional Office for Africa, Brazzaville, Republic of Congo, 6 School of Public Health and Community Medicine, University of New South Wales, Sydney, Australia, 7 The George Institute for Global Health, New Delhi, India

‡ Equal senior authors.
* atesema@georgeinstitute.org.au, azeb18@gmail.com

**Data Availability Statement:** All relevant data are within the manuscript and its Supporting

## Abstract

### Objective

In Africa, mortality due to non-communicable diseases (NCDs) is projected to overtake the combined mortality from communicable, maternal, neonatal, and nutritional diseases by 2030. To address this growing NCD burden, primary health care (PHC) systems will require substantial re-orientation. In this study, we reviewed the progress of African countries towards integrating essential NCD services into PHC.

### Methods

A review of World Health Organization (WHO) reports was conducted for all 47 countries in the WHO African Region. To report each country's progress, we used an a priori framework developed by the WHO regional office for Africa (AFRO). Twelve indicators were used to measure countries' progress. The proportion of countries meeting each indicator was tabulated using a heat map. Correlation between country income status and attainment of each indicator was also assessed.

### Findings

No country met all the recommended indicators to integrate NCD services into PHC and seven countries met none of the indicators. Few countries (30%) had nationally approved guidelines for NCD management and very few reported availabilities of all essential NCD

Information files can be accessed from the World Health Organization database.

**Funding:** The UNSW Scientia Scholarship program supports AT and WA. SA was supported by the Australian National Health and Medical Research Council (NHMRC) through an Overseas Early Career Fellowship (APP1139631). RJ is supported by the Australian National Heart Foundation (APP 102059) and a UNSW Scientia Fellowship. DP is support by NHMRC career Development Fellowship, Level 2 and Australia National Heart Foundation Future Leader Fellow. The funders has no role in stusy design, data collection and analysis, decision to publish, or preparation of the manuscript.

**Competing interests:** The authors have declared that no competing interests exist.

medicines (13%) and technologies (11%) in PHC facilities. There was no overall correlation between a country's GDP per capita and the aggregate of targets being met (rho = 0.23; $P$ = .12). There was, however, a modestly negative correlation between out-of-pocket expenditure and overall country progress (rho = -0.58; $P$ < .001).

## Conclusion

Progress by AFRO Member States in integrating NCD care into PHC is variable across the region. Enhanced government commitment and judicious resource allocation to prioritize NCDs are needed. Particular areas of focus include increasing the uptake of simplified guidelines for NCDs; increasing workforce capacity to manage NCDs; and removing access barriers to essential medicines and basic diagnostic technologies.

## Introduction

Non-communicable diseases (NCDs) were responsible for an estimated 40.5 million deaths (71% of all deaths) in 2016 [1], of which 78% occurred in low and middle-income countries (LMICs). In Africa, the age standardised disability-adjusted life-years (DALYs) due to NCDs is almost equivalent to the combined DALYs from communicable, maternal, neonatal, and nutritional (CMNN) conditions in 2017 [2], and is projected to exceed deaths due to CMNN conditions by 2030 [3, 4]. Health systems in Africa are struggling to address the growing NCD burden. Inaccessible medicines and technologies [5], high out-of-pocket (OOP) costs [6], limited integration with primary health care (PHC) [7] and low density of the health workforce are barriers to an adequate health system response to NCDs [7]. Given the limited fiscal space available to health system planners to address these issues, there is need for cost-effective and equitable programs delivered through PHC [7–10].

World leaders made commitments to address NCDs at the UN General Assembly high-level meeting in 2011 and reaffirmed their actions in 2014 and 2018. In 2016, NCDs were included in the Sustainable Development Goal target 3.4 "by 2030, reduce by one third premature mortality from NCDs through prevention and treatment and promote mental health and well-being" [11, 12]. The World Health Organization (WHO) Global action plan for the control of NCD 2013–2020 recommended that countries strengthen their health systems and address NCDs through people-centred PHC and universal health coverage (UHC) [13]. This action reaffirmed in the third UN General Assembly high-level meeting for NCDs in 2018 [14]. These recommendations including 'Best Buys' (guidance on the most cost-effective interventions to invest in) [15, 16] and WHO Package of Essential Noncommunicable disease interventions (WHO PEN), (cost-effective interventions for the early detection and management of NCDs) have been successfully implemented by many countries [8, 17–19].

Despite encouraging evidence of the impact of these interventions, and high-level commitment, many countries' policy responses to NCD control are limited. Even when polices are in place, there is variation in implementation [18–22]. Previous studies conducted to track Africa's progress in achieving NCD targets found that more than half of the countries had not achieved the interim targets [23] and progress was slow [8, 24].

In response to this, the 67th Session of the WHO African regional committee adopted a framework for integrating essential NCD services at the PHC level [7, 23] which provides targeted guidance to Member States. Four targets have been set to be achieved by 2030: (1) adapt

and use WHO PEN; (2) more than 80% of the PHC workforce should receive formal training in NCD management; (3) have the essential medicines and basic technologies needed for NCD management available at PHC facilities; and (4) have systems for collecting mortality data routinely [7].

The aims of this study were to: (1) review African regional progress towards integration of essential NCDs services at the PHC level; and (2) to examine associations between financial indicators (country-level income status and out of pocket costs) and provision of essential NCD services and NCD achievements being met.

## Methods

A review of WHO documents from all 47 countries in the WHO African Region was conducted. The review procedure is detailed in the below sub-sections.

### Defining indicators

Indicators of progress were derived from the four WHO African regional committee targets for integrating essential NCD services for the early detection and management of CVD, diabetes, chronic respiratory diseases and cancer in PHC. Ten of twelve indicators were derived from the four regional targets which covered: (1) use of the WHO guidance documents for NCD management within PHC services (2) PHC workforce policies, particularly the training of the non-physician healthcare workers; (3) provision of essential medicines and equipment relevant to NCD management at the PHC level and; (4) information management of NCD data. Two additional indicators, healthcare financing for NCD care and the commitment of governments to govern and manage NCD program, were included to more holistically assess progress. The targets, indicators, definitions and data sources are summarized in Table 1.

### Identification of relevant documents and scoring

For indicator data sources, we extracted information mainly from WHO reports [7, 25, 26]. The primary data sources included the WHO 2020 NCD progress monitoring report [25]; 2018 WHO NCD country profiles [26] and the report from the 67th session of WHO Regional Committee for Africa [7]. The WHO document repository for NCDs and Global Health Expenditure database were accessed for financing data [27, 28]. In addition, we worked with the WHO regional office for Africa to obtain data that were not yet publicly reported (Table 1). We did not have rigid inclusion and exclusion criteria; however, documents that did not provide a national overview were excluded.

In order to score progress, we used definitions in line with the WHO NCD progress monitoring report 2020 (2nd, 3rd, 9th and 10th indicators). That is; 'fully achieved' when the countries achieved all the required standards; 'partially achieved' when they had incomplete achievements and 'not achieved' where there was no evidence of progress [25]. For indicators in the WHO 2018 country profile [26] and indicators from other sources, we categorized the data in line with the 2020 WHO progress monitoring report. For healthcare financing, we used OOP expenditure as the primary indicator and classified the countries based on the WHO global health spending report [29, 30] (Table 1). We accorded a value of three points for each fully achieved indicator, two for partially achieved indicator, one for indicator that had not been achieved, and zero for which there were no data available. We generated aggregate scores for each country with a maximum score of 36 points indicating all 12 indicators were fully met.

Data were extracted and mapped against the 12 indicators (Table 1). Three researchers (AGT, WSA and CB) extracted the data and entered information into a Microsoft Excel spreadsheet. Each researcher extracted data separately and later crosschecked each other's

**Table 1. Summary of the data extraction indicators, measurement, and data source for mapping the progress of African countries to integrate essential NCDs services, using the WHO regional committee for Africa targets, 2020.**

| Regional target | Indicator | Measurement used (including in the heat map) | Data source |
|---|---|---|---|
| **Target 1:** Member States adapted and are using the WHO PEN | I. Country have national guidelines/protocols/ standards for the management of four NCDs through a primary care approach approved by government | 3 = Fully achieved: if document exist for all four NCDs | • 2020 WHO progress monitoring report, indicator 9 of the report https://apps.who.int/iris/bitstream/handle/10665/330805/9789240000490-eng.pdf?sequence=1&isAllowed=y |
| | | 2 = Partially achieved: if document exist for two of the four NCDs | |
| | | 1 = Not achieved: no documentation available for any NCD | |
| | | 0 = No data found/not known | |
| | II. Country have adapted and are using the WHO PEN | 3 = Fully achieved: if the country adapts and uses WHO PEN | • Data was obtained from the WHO regional office for Africa |
| | | 2 = Partially achieved: if the country adapts and pilots WHO PEN to develop their strategy | WHO 67th regional committee for Africa report https://www.afro.who.int/sites/default/files/2017-08/AFR-RC67-12%20Regional%20framework%20to%20integrate%20NCDs%20in%20PHC.pdf |
| | | 1 = Not achieved: if the country did not adapt or use WHO PEN | |
| | | 0 = No data found/not known | |
| **Target 2:** Training of health care workers for managing NCDs at PHC level | III. Country has a strategic plan to train PHC workers on management of NCDs | 3 = Yes: country has trained or plans to train PHC workers | • Country strategy was accessed from WHO's NCDs Document Repository https://extranet.who.int/ncdccs/documents/Db |
| | | 1 = No: Data unavailable- if the country doesn't have a strategy at all, or documents don't indicate the PHC health worker training (this include document written in other languages) | • Specific country reports and strategic documents |
| **Target 3:** Member States have essential medicines and basic technologies for NCDs in PHC facilities | IV. National essential medicines and basic technology list for NCDs (PHC level, if available) | 3 = National essential medicine and basic technologies list is available, and also country have a list for PHC | • Data was extracted from WHO National medicines list/formulation/standard treatment guideline portal https://www.who.int/selection_medicines/country_lists/en/ |
| | | 2 = National essential medicine/basic technologies list is available | • National government websites, journal articles and newspaper articles were searched |
| | | 1 = if the country didn't have an essential medicine and basic technologies list or no list could be identified | |
| | V. Number of essential NCD medicines reported as "generally available" in primary care facilities of the public health sector ("generally available" were described if medicine/technology were available in 50% or more of pharmacies/facilities) | • Number of essential medicines available out of 10 | • 2018 WHO NCDs country profile https://www.who.int/nmh/publications/ncd-profiles-2018/en/ |
| | | The 10 essential NCD medicines include; Aspirins, Statins, Angiotensin-converting enzyme inhibitors, Thiazide diuretics, Long-acting calcium channel blockers, Beta-blockers, Insulin, Metformin, Bronchodilators, and Steroid inhalants | |
| | | For this study, data was categorized; | |
| | | 3 = All the 10 NCDs medicines were available | |
| | | 2 = 5–9 NCDs medicines were available | |
| | | 1 = Less than 5 NCDs medicines were available | |
| | | 0 = No data available | |
| | VI. Number of essential NCD technologies reported as "generally available" in primary care facilities of the public health sector | • Number of essential technologies available out of 6 | • 2018 WHO NCDs country profile https://www.who.int/nmh/publications/ncd-profiles-2018/en/ |
| | | The 6 basic technologies include; Blood pressure measurement devices, weighing scales, height measuring equipment, blood sugar and blood cholesterol measurement devices with strips, and urine strips for albumin assay | |
| | | For this study, data was categorized; | |
| | | 3 = All the 6 technologies were available | |
| | | 2 = 3–5 technologies were available | |
| | | 1 = <3 technologies were available | |
| | | 0 = No data available | |
| | VII. Proportion of PHC centres reported to offer CVD risk stratification | • Percentage of PHC centres who offer the service | • 2018 WHO NCDs country profile https://www.who.int/nmh/publications/ncd-profiles-2018/en/ |
| | | For this study, data was categorized; | |
| | | 3 = countries with > 50% facilities provide service | |
| | | 2 = countries with < 25% facilities provide service | |
| | | 1 = None/service not available | |
| | | 0 = No data available/ don't know | |
| | VIII. Member State has provision of drug therapy, including glycaemic control, and counselling for eligible persons at high risk to prevent heart attacks and strokes, with emphasis on the primary care level | 3 = Fully achieved: if the country reports that more than 50% of PHC facilities are offering the services | • 2020 WHO NCD progress monitoring, indicator 10 https://apps.who.int/iris/bitstream/handle/10665/330805/9789240000490-eng.pdf?sequence=1&isAllowed=y |
| | | 2 = Partially achieved: if the country reports that between 25% to 50% of PHC facilities are offering the services | |
| | | 1 = Not achieved: if the country did report or don't offer the service | |
| | | 0 = No data found/ not known | |

*(Continued)*

**Table 1.** (Continued)

| Regional target | Indicator | Measurement used (including in the heat map) | Data source |
|---|---|---|---|
| **Target 4:** To strengthen and integrate NCD surveillance systems into health management information systems | IX. Member State has a functioning system for generating reliable cause-specific mortality data on a routine basis | 3 = Fully achieved- if country meets all the criteria for reliable cause-specific mortality data (the criteria are: 70% usable; at least five years of cause-of-death data reported to the WHO in the last 10 years and recent data reported for WHO within 5 years) | • 2020 WHO NCD progress monitoring report- indicator 2 https://apps.who.int/iris/bitstream/handle/10665/330805/9789240000490-eng.pdf?sequence=1&isAllowed=y |
| | | 2 = Partially achieved- if the country does not meet all the criteria but has submitted some vital registration data to WHO | |
| | | 1 = Not achieved otherwise | |
| | | 0 = No data found/not known | |
| | X. Has a STEPS survey or a comprehensive health examination survey every 5 years | 3 = Fully achieved- if country answer responds "Yes" to each of the risk factors covered in the STEP survey; the country must indicate that the last survey was conducted in the past 5 years and country must also provide the needed supporting documentation | • 2020 WHO progress monitoring report, indicator 3 (List of risk factors are available in the appendix 1 of the report) https://apps.who.int/iris/bitstream/handle/10665/330805/9789240000490-eng.pdf?sequence=1&isAllowed=y |
| | | 2 = Partially achieved- if the country achieved some of but not all the risk factors listed in the STEP survey, or the surveys were conducted more than 5 years ago but less than 10 years ago. | |
| | | 1 = Not achieved otherwise | |
| | | 0 = No data found/not know | |
| **Additional Indicators** | | | |
| Health care financing | XI. Out-of-pocket (OOP) expenditure as % of current health expenditure/2016 | The proportion of OOP expenditure as % of Current Health Expenditure/2016 | • WHO Global Health Expenditure Database http://apps.who.int/nha/database/DocumentationCentre/Index/en https://apps.who.int/iris/bitstream/handle/10665/276728/WHO-HIS-HGF-HF-WorkingPaper-18.3-eng.pdf?ua=1 |
| | | Spending categorised as: | |
| | | 3 = OOP < 20% | |
| | | 2 = OOP = 21–37% | |
| | | 1 = OOP ≥ 38% (above the regional average) | |
| | | 0 = No data available/not know | |
| Leadership | XII. Existence of an operational Unit, branch, or department in ministry of Health with responsibility for NCDs | 3 = The country has a unit or other equivalent body in the country | • WHO 2017 report from Global Health Observatory data repository https://apps.who.int/gho/data/view.main.2472 Effective date: 2018-03-13 |
| | | 1 = No body available | |

works to maintain the quality of the data. The research team met regularly to discuss the data extraction and management process. Furthermore, government strategic documents were reviewed qualitatively to supplement the numeric data.

Once the data extraction process was complete, a draft report was sent to the WHO Africa regional office team to provide supplementary information, focussing on those countries with limited data. This also validated the accuracy of the data extracted for other countries. The WHO Africa regional office team contacted the respective Ministry of Health/ WHO focal person to verify the availability of the data. To facilitate the interpretation of findings, we also conducted targeted searches of country reports and strategic documents, with particular attention to indicators for which where country-level data were limited or missing in WHO reports.

## Data analysis

Scores were tabulated and displayed using a heat map to compare each country's achievement against various indicators. To examine correlation between financial variables and attainment of indicators, we used Gross Domestic Product per capita (GDP per capita) and OOP costs and assessed correlation with overall aggregate scores, NCD medicine scores and NCD technologies. Spearman's rank correlation coefficient (rho) was calculated and the size of the correlation coefficient was interpreted based on the classification stated by Mukaka, where a value approaching 1 indicates high correlation, and a value less than 0.3 suggests low or negligible correlation [31]. Data analysis was done using IBM SPSS statistical software, version 25.

### Ethics approval

This study used open access documents; therefore, no ethical issues involved.

## Results

### Characteristics of the countries

Data from all the 47 countries in the WHO African region were included. Twenty-six countries (55%) were classified as low-income countries (LICs),13 (28%)lower-middle-income countries (LMICs), 7(15%) upper-middle-income countries (UMICs) and one was classified as a high-income country (Seychelles) [32]. The estimated proportional mortality due to NCDs varied from 26% (Central African Republic) to 89% (Mauritius) of all deaths [26]. Upper-middle and HICs [Mauritius (89%), Seychelles (81%) and Algeria (76%)] had the highest proportion of NCD deaths [26]. The main findings based on the regional targets are summarised below and in Table 2.

### Target one: Adapt and use WHO guidance documents for NCD management within PHC services

**Indicator I: National guidelines/protocols/standards for the management of major NCDs through a primary care approach.**   Among the 47 countries, 14(30%) had PHC guidelines for the management of NCDs, while 17 (36%) countries had not achieved this target (Table 3).

**Indicator II: Adaptation and use of WHO Package of Essential NCDs (PEN) interventions.**   By December 2018, 13 (28%) countries had adapted and were using the WHO PEN (Table 3). In addition, 14 country representatives were trained on the implementation of the WHO PEN interventions [33].

### Target two: Train the PHC workforce in managing NCDs

**Indicator III. Strategic plan to train the PHC workforce to manage NCDs.**   Even though we could not access the number of health workers trained in each country, 17 (36%) countries reported that either health workers have been trained or have a national strategy that includes a plan to train PHC workers for NCDs management. For the remaining 30 countries, either their NCDs strategic plan could not be accessed or the documents did not provide information about training.

### Target three: Availability of essential medicines and technologies for NCD management in PHC facilities

**Indicator IV: List of essential medicines and technologies for NCDs management nationally (PHC facilities, if any).**   Of the 40 (85%) countries that had essential medicines list at the national level, three (7%) countries (Angola, Sierra Leone and South Africa) had specific essential medicine lists for PHC facilities (Table 2).

**Indicators V & VI: Essential NCD medicines and technologies in PHC facilities.**   Six countries (13%) reported having all ten recommended essential NCD-related medicines on their list, and five countries (11%) reported having the six NCD essential technologies "generally available" in their PHC facilities (Table 3). Only two countries [Cabo Verde and Mauritius—both UMICs] reported having all ten essential NCD medicines and six essential NCD technologies. Angola, Equatorial Guinea, and Gabon (UMICs), and Niger and South Sudan (both LICs) did not report availability of essential NCD medicines in PHC facilities (Tables 2 and 3).

Table 2. Heat map showing the progress of countries against the WHO regional committee for Africa target to integrate NCDs services in the PHC level, 2020.

| Country Name | Indicators | | | | | | | | | | | |
|---|---|---|---|---|---|---|---|---|---|---|---|---|
| | I. NCD guideline in PHC | II. WHO PEN | III. HRH trained | IV. National list of EM and technology | V. Generally available medicine/10 | VI. Generally available technologies/6 | VII. CVD risk stratification @PHC | VIII. Country has drug therapy @PHC | IX. Cause-specific mortality data | X. Has STEP survey | XI. Out-of-pocket expenditure | XII. Leadership |
| Algeria | 2 | 1 | 1 | 2 | 2 | 3 | 0 | 1 | 1 | 2 | 2 | 3 |
| Angola | 0 | 1 | 3 | 3 | 1 | 2 | 1 | 2 | 1 | 2 | 2 | 3 |
| Benin | 3 | 3 | 1 | 2 | 1 | 1 | 2 | 2 | 1 | 2 | 1 | 3 |
| Botswana | 3 | 3 | 1 | 2 | 2 | 2 | 0 | 2 | 1 | 3 | 3 | 1 |
| Burkina Faso | 3 | 3 | 1 | 2 | 1 | 2 | 1 | 2 | 1 | 2 | 2 | 3 |
| Burundi | 2 | 1 | 1 | 2 | 1 | 2 | 1 | 2 | 1 | 1 | 2 | 1 |
| Cabo Verde | 1 | 2 | 1 | 2 | 3 | 3 | 3 | 3 | 2 | 1 | 2 | 3 |
| Cameroon | 1 | 1 | 1 | 2 | 1 | 2 | 2 | 2 | 1 | 1 | 1 | 3 |
| Central African Republic | 2 | 1 | 1 | 2 | 2 | 2 | 2 | 2 | 1 | 2 | 1 | 1 |
| Chad | 1 | 1 | 1 | 2 | 1 | 2 | 2 | 2 | 1 | 1 | 1 | 1 |
| Comoros | 0 | 1 | 1 | 1 | 2 | 2 | 3 | 2 | 1 | 2 | 1 | 1 |
| Congo | 2 | 1 | 1 | 2 | 1 | 2 | 2 | 2 | 1 | 1 | 1 | 1 |
| Côte d'Ivoire | 1 | 3 | 1 | 2 | 1 | 2 | 2 | 2 | 1 | 2 | 1 | 3 |
| D.R Congo | 2 | 1 | 1 | 2 | 2 | 1 | 1 | 2 | 1 | 1 | 2 | 1 |
| Equatorial Guinea | 1 | 1 | 1 | 1 | 1 | 1 | 2 | 2 | 1 | 1 | 1 | 1 |
| Eritrea | 1 | 3 | 3 | 2 | 1 | 2 | 2 | 2 | 1 | 2 | 1 | 1 |
| Eswatini/Swaziland | 1 | 3 | 3 | 2 | 3 | 2 | 2 | 2 | 1 | 3 | 3 | 3 |
| Ethiopia | 3 | 3 | 3 | 2 | 1 | 2 | 2 | 2 | 1 | 3 | 2 | 3 |
| Gabon | 1 | 1 | 1 | 2 | 1 | 2 | 2 | 2 | 1 | 1 | 2 | 1 |
| Gambia | 1 | 1 | 3 | 1 | 1 | 2 | 0 | 1 | 1 | 2 | 2 | 1 |
| Ghana | 3 | 2 | 3 | 2 | 2 | 2 | 1 | 0 | 1 | 2 | 1 | 1 |
| Guinea | 2 | 3 | 1 | 2 | 2 | 2 | 2 | 2 | 1 | 1 | 1 | 1 |
| Guinea-Bissau | 1 | 1 | 1 | 1 | 1 | 2 | 1 | 2 | 1 | 1 | 2 | 3 |
| Kenya | 3 | 1 | 3 | 2 | 2 | 2 | 2 | 2 | 1 | 3 | 2 | 3 |
| Lesotho | 3 | 3 | 3 | 2 | 2 | 2 | 0 | 2 | 1 | 2 | 3 | 1 |
| Liberia | 2 | 1 | 1 | 2 | 1 | 2 | 1 | 2 | 1 | 2 | 1 | 1 |
| Madagascar | 3 | 1 | 1 | 2 | 1 | 1 | 1 | 2 | 1 | 1 | 2 | 3 |
| Malawi | 3 | 3 | 3 | 2 | 2 | 1 | 1 | 2 | 1 | 2 | 3 | 3 |
| Mali | 1 | 1 | 1 | 2 | 2 | 3 | 0 | 1 | 1 | 2 | 2 | 1 |
| Mauritania | 1 | 1 | 1 | 2 | 2 | 1 | 2 | 2 | 1 | 1 | 1 | 1 |
| Mauritius | 2 | 1 | 1 | 1 | 3 | 3 | 1 | 2 | 3 | 2 | 1 | 3 |
| Mozambique | 2 | 1 | 1 | 2 | 2 | 2 | 2 | 2 | 1 | 2 | 3 | 3 |
| Namibia | 1 | 1 | 1 | 2 | 2 | 2 | 1 | 2 | 1 | 2 | 3 | 3 |
| Niger | 1 | 1 | 1 | 2 | 1 | 2 | 1 | 2 | 1 | 1 | 1 | 1 |
| Nigeria | 1 | 1 | 3 | 2 | 1 | 1 | 1 | 2 | 1 | 1 | 1 | 1 |
| Rwanda | 3 | 1 | 1 | 2 | 3 | 2 | 2 | 1 | 1 | 2 | 3 | 3 |
| Sao Tome and Principe | 2 | 1 | 1 | 1 | 2 | 2 | 1 | 2 | 1 | 2 | 3 | 3 |
| Senegal | 3 | 1 | 1 | 2 | 2 | 3 | 1 | 2 | 1 | 3 | 1 | 1 |
| Seychelles | 2 | 2 | 3 | 2 | 3 | 2 | 2 | 3 | 3 | 2 | 3 | 3 |
| Sierra Leone | 1 | 3 | 3 | 3 | 1 | 2 | 2 | 2 | 1 | 1 | 1 | 1 |
| South Africa | 2 | 1 | 3 | 3 | 3 | 2 | 0 | 1 | 3 | 2 | 3 | 3 |

(Continued)

**Table 2.** (Continued)

| Country Name | Indicators | | | | | | | | | | | |
| --- | --- | --- | --- | --- | --- | --- | --- | --- | --- | --- | --- | --- |
| | I. NCD guideline in PHC | II. WHO PEN | III. HRH trained | IV. National list of EM and technology | V. Generally available medicine/ 10 | VI. Generally available technologies/ 6 | VII. CVD risk stratification @PHC | VIII. Country has drug therapy @PHC | IX. Cause-specific mortality data | X. Has STEP survey | XI. Out-of-pocket expenditure | XII. Leadership |
| South Sudan | 1 | 1 | 1 | 1 | 1 | 2 | 1 | 2 | 1 | 1 | 0 | 1 |
| Togo | 2 | 3 | 1 | 2 | 2 | 2 | 2 | 2 | 1 | 2 | 1 | 1 |
| U.R Tanzania | 3 | 1 | 3 | 2 | 2 | 2 | 2 | 2 | 1 | 2 | 2 | 1 |
| Uganda | 3 | 2 | 3 | 2 | 2 | 2 | 2 | 2 | 1 | 3 | 1 | 3 |
| Zambia | 3 | 1 | 3 | 2 | 2 | 2 | 2 | 2 | 1 | 2 | 3 | 3 |
| Zimbabwe | 2 | 1 | 1 | 2 | 1 | 2 | 1 | 2 | 1 | 1 | 2 | 1 |

3 = Full/good achievement for the indicator mentioned

2 = Partial achievement for the indicator mentioned

1 = No achievement for the indicator mentioned

0 = no data available during extraction of documents

**Indicator VII: Provision of drug therapy, including glycaemic control, and counselling for people at high risk to prevent heart attacks and strokes, at PHC facilities.** This indicator was fully achieved in Cabo Verde (LMIC) and Seychelles(HIC) and partially achieved in 40 (80%)countries(Tables 2 and 3).

**Indicator VIII: Proportion of PHC centres reported as offering cardio-vascular diseases risk stratification.** Among the 47 countries, Carbo-Verde and Comoros reported having Cardiovascular Diseases (CVD) risk stratification in over 50% of its PHC facilities. In 18 (38%) countries there was no risk stratification conducted in PHC facilities (Tables 2 and 3).

## Target four: Health information systems

**Indicators IX & X: Functioning system for generating reliable cause-specific mortality and morbidity data using either WHO's STEPwise approach to surveillance (STEPS) or a comprehensive health survey every 5 years.** Seychelles (HIC), Mauritius (MIC) and South African(UMIC) reported fully established, reliable and routine systems to generate cause-specific NCDs mortality data. Cabo Verde reported partial achievement, and the remaining 43 (92%) countries reported no system for routine collection of NCD mortality data. In addition, six (11%) countries Botswana, Kenya, and Lesotho (MIC), and Uganda (LIC) conduct STEPS or comprehensive health surveys every five years (Tables 2 and 3).

## Additional indicators

**Health system financing.** *Indicator XI: Out-of-pocket expenditure as percentage of current health expenditure.* Eleven (23%) countries (one HIC, seven MICs, and three LICs) had low (≤20%) OOP expenditure as a proportion of total health expenditure in 2016. Of these, six countries: Seychelles (HIC), Botswana, Namibia and South Africa (UMICs), and Rwanda and Mozambique (LICs) reported <10% OOP expenditure on health. Twenty (43%) countries have above the regional average of OOP spending with ≥38% [the average OOP spending for

Table 3. Percentage of achievement for various indicators: Mapping African countries' progress toward integration of NCDs service at the PHC level, 2020.

| Indicator | Number of country (n = 47) | Percent (%) |
|---|---|---|
| *Availability of guidelines for the management of major NCDs at the PHC level* | | |
| Fully achieved | 14 | 30 |
| Partially achieved | 14 | 30 |
| Not achieved | 17 | 36 |
| No available data | 2 | 4 |
| **Country adopted and use WHO PEN** | | |
| Adopted and using WHO PEN | 13 | 28 |
| List WHO PEN as a strategy/ pilot test WHO PEN in specific area of the country | 3 | 6 |
| Have not adopted WHO PEN | 31 | 66 |
| **Training of primary healthcare workers for NCDs management** | | |
| Country has trained or plans to train primary healthcare workers | 17 | 36 |
| No available data /document written in other language | 30 | 64 |
| **Essential medicine LIST available nationally and at the PHC level** | | |
| Essential medicine available nationally and at the PHC level | 3 | 7 |
| Essential medicine available at the national level only | 40 | 85 |
| No available data | 4 | 8 |
| **Number of essential medicines available** | | |
| <5 NCD medicines | 22 | 48 |
| 5–9 medicines | 19 | 40 |
| All the 10 medicines | 6 | 12 |
| **Number of essential NCDs technology available** | | |
| <3 technologies | 7 | 15 |
| 3–5 technologies | 35 | 74 |
| All the 6 technologies | 5 | 11 |
| **Proportion of primary health care centres reported as offering CVD risk stratification** | | |
| < 25% (including one country which provide in25-50% of its facilities) | 21 | 45 |
| > 50% | 2 | 4 |
| None/service not available | 18 | 38 |
| Don't know (DK)/data not found | 6 | 13 |
| **Provision of drug therapy, including glycaemic control, and counselling for eligible persons at high risk to prevent heart attacks and strokes, with emphasis on the primary care level** | | |
| Fully achieved | 2 | 4 |
| Partially achieved | 40 | 85 |
| Not achieved | 4 | 9 |
| No data | 1 | 2 |
| *Member State has a functioning system for generating reliable cause-specific mortality data* | | |
| Fully achieved | 3 | 6 |
| Partially achieved | 1 | 2 |
| Not achieved | 43 | 92 |
| *having STEPS survey or a comprehensive health examination survey every 5 years* | | |
| Fully achieved | 6 | 13 |
| Partially achieved | 24 | 51 |
| Not achieved | 17 | 36 |
| **Out-of-pocket expenditure as percentage of current health expenditure** | | |
| ≤ 20% OOP expenditure | 11 | 23 |

(*Continued*)

**Table 3.** (Continued)

| Indicator | Number of country (n = 47) | Percent (%) |
|---|---|---|
| 21% - 37% OOP expenditure | 15 | 32 |
| ≥ 38% OOP expenditure | 20 | 43 |
| No available data | 1 | 2 |
| **Existence of an operational unit, branch, or department in Ministry of Health with responsibility for NCDs program** | | |
| Yes | 22 | 47 |
| No | 25 | 53 |

African region is 37% [30]]. Cameroon, Comoros and Equatorial Guinea (MICs) reported the highest proportions of OOP expenditure.

**Leadership and governance.** *Indicator XII*: *Existence of an operational department in Ministry of Health responsible for NCDs program*. The establishment of new or strengthening existing operational unit or department responsible for planning, implementation and evaluation of NCD programs is very crucial in fostering effective integration of the service at all levels. Twenty-two (47%) countries have an operational unit or department responsible to provide leadership and guidance for the implementation of NCDs activities (Table 3).

## Progress scores and correlation with financial indicators

Cabo Verde, Ethiopia, Eswatini, Kenya, Malawi, Rwanda, Seychelles, South Africa, Uganda, and Zambia scored ≥25 out of a maximum of 36 points. By contrast, Chad, Democratic Republic of Congo, Equatorial Guinea, Gambia, Mauritania, Niger and South Sudan had the lowest scores with no targets fully achieved (Table 2). There was no overall correlation between countries' GDP per capita and the aggregate score of each country's achievement (rho = 0.23; *P* = .12). However, a modest negative correlation was observed between countries' OOP expenditure and their aggregate achievement score. This showed, countries with relatively lower OOP expenditure had higher aggregate scores (rho = -0.58; *P* < *.001*) (Table 4).

With respect to NCD medicines and technologies, there was no correlation between GDP per capita and the availability of NCD medicines (rho = 0.31; *P* = .03) however, there was a positive correlation between GDP per capita and essential technologies (rho = 0.37; *P* = .009) (Table 4). Also, OOP expenditure displayed a moderate negative correlation with the number of essential NCD medicines available in the country. Countries with a relatively higher number of essential NCD medicines had lower OOP expenditure (r = -0.51; *P* < .001) (Table 4).

## Discussion

We assessed progress of countries towards PHC integration of essential NCD services in the WHO African region and identified areas of relative strength and weakness in achieving the regional targets. Although no country in the region had met all the indicators assessed, some demonstrated progress. Relatively higher achievements were recorded from Cabo Verde (LMIC), Ethiopia (LIC), Eswatini (LMIC), Malawi (LIC), Rwanda (LIC), Seychelles (HIC) and South Africa (UMIC). The lowest achievement scores were observed in South Sudan (LIC), Equatorial Guinea (UMIC), and Niger (LIC). Overall there was a mixed picture with associations between financial indicators and progress against targets. Progress was not directly correlated with the income level of countries; however, for a specific indicator (access to essential NCD technologies) positive correlation was displayed with the country's income level. For

**Table 4. Correlation coefficient results for selected variables, African countries' progress toward integration of NCDs service at the PHC level, 2020.**

|  | Number of essential NCDs medicine available in the country | | Number of essential NCDs technology available in the country | | Aggregate score | |
|---|---|---|---|---|---|---|
|  | Spearman's rho, | p-value | Spearman's rho, | P-value | Spearman's rho, | P-value |
| Gross Domestic Product (GDP) per capital, 2017 | 0.31 | 0.03 | 0.37 | 0.009** | 0.21 | 0.15 |
| Out-of-Pocket (OOP) expenditure | -0.51 | 0.0001** | -0.25 | 0.09 | -0.58 | 0.0001** |

**Correlation is significant at p-value < 0.01 level (2-tailed)

OOP expenditure there was a modest negative correlation with the country's progress and with access to essential NCD medicines, but not with access to essential NCD technologies. The findings suggest that health system drivers to make progress in NCD targets are more complex than simply the wealth of the nation.

Variable progress was evident across all target areas. With regard to use of NCD guidelines, only 30% of countries have nationally approved NCD plans for implementation in PHC, a slight improvement from 17% in 2014 [23] and 26% from 2017 [24]. In 2017, among the 31 countries who had a national NCD strategic plan, only 17 were operational [33]. We also found few countries (47%) reported a distinct department for NCD programs. In 2017, only two-thirds (77%) of countries had one full-time professional staff member in the ministry representing a decline from previous years 83% (2010), 93% (2013) and 100% in 2015 [34].

Only 13 countries in the region have implemented WHO PEN interventions at the PHC level, indicating that activities need to be scaled-up [7]. Effective adaptation and implementation of WHO PEN guideline is a complex undertaking. Studies in Ghana and Zambia have identified critical capacity gaps in PHC facilities, particularly workforce capacity, health system infrastructure and health information systems [35, 36]. Addressing these gaps requires a multi-sectoral response with a commitment to invest in PHC. Priority areas identified in this study include: (1) the need for substantial resource commitments to strengthen and expand the workforce; (2) adequate financing and supply chains to ensure access to essential medicines and diagnostic technologies, and (3) investment in robust health information systems [35–37].

Most countries (64%) lack national plans to train PHC workers for NCD management, which is well short of achieving the target of having at least 50% of PHC workers trained in NCD management by 2020. This is exacerbated by intra-country variation with 75% of countries experiencing both shortage and mal-distribution of the workforce [7]. Task shifting to non-physician healthcare workers (typically nurses and community health workers) is a well-established strategy and valuable lessons can be learnt from successful programmes such as HIV care and maternal and child health [16, 38–40].

Similar to previous studies [41–44], many countries reported limited availability of essential NCD medicines and technologies. Only Cabo Verde and Mauritius report availability of all ten essential NCD medicines and six essential NCD technologies in their PHC facilities. Cabo Verde has achieved these targets through strong political commitment to strengthen PHC services as part of their UHC agenda [45]. The limited availability of affordable medicines and diagnostic tests is a complex challenge in sub-Saharan Africa [43]. Minimal competition among few numbers of suppliers in LMICs affects medicine supply and cost. A 2019 report showed that buyers in LMICs pay up-to 20–30 times the minimum international reference price for generic medicines [46]. Generating sufficient fiscal space to subsidise the costs of these medicines and avoid passing on the costs to end users is essential.

Despite global advocacy for strengthening health information systems, such systems remain under-developed in the region. Most countries (92%) lack sufficient systems for generating reliable NCD mortality data. With the latest 2020 WHO progress report, six countries (Botswana, Ethiopia, Eswatini, Kenya, Lesotho and Uganda) have made progress with conducting and repeating STEPS regularly. Only two countries, Ethiopia and Eswatini have made progress comparing to the 2017 progress report [24]. However, the 2018 WHO Africa Region Secretariat reports a decreasing trend of STEPS implementation in countries from 33 in 2003, to 15 in 2015 [8]. Establishing robust civil registration and vital statistics (CRVS) systems by allocating resources and training staff to generate disease specific, age and sex disaggregated data, and integrating NCD data within existing Demography Heath Survey are priorities for improving NCD surveillance [34, 47, 48]. Several WHO programmes are underway to address some of these priorities, providing technical and financial support to improve CRVS systems in Ghana, Malawi, Rwanda and Tanzania [49].

We observed countries with high OOP expenditure have lower availability of NCD medicines in the public sector. And, countries with higher GDP per capita, had slightly higher availability of NCD technologies. There were exceptions with some LIC and LMICs like Cabo Verde, Mali and Senegal reporting the availability of all essential NCD technologies. This suggests that several other factors are at play beyond resource availability such as the country's commitment and priority setting. A study in 151 countries, suggested that NCD policy implementation is not necessarily expensive, and found that many MICs outperform HICs [50].

There are limitations to this review. The review mainly used data from the WHO reports, and we are constrained by the limitations in those reports. These reports are sourced from each country and this could bias the findings. To improve the quality of the data, WHO requested countries to provide supporting documentation pertaining to the relevant indicator/s. We found limited information to assess regional variation on progress and limited information on workforce training with only the ability to review plans for training rather than reports on actual implementation. Except for a few countries (e.g. Ethiopia and Nigeria) where the researchers had access to local networks to verify actual progress, we were not able to do that for all countries. This was mitigated by consulting with the WHO regional office and reviewing peer-reviewed journals to corroborate findings. Going forward to address these limitations we recommend: (1) independent monitoring mechanisms to assess implementation of NCDs interventions; (2) develop systems that enable reporting of NCD targets that are disaggregated by region; and (3) expand evaluation research in the field taking into consideration regional complexity.

## Conclusion

This review demonstrates the need for intensified efforts to integrate essential NCD services at the PHC level in the WHO AFRO Member States. Priority areas include: (1) reorientation of PHC systems to better integrate NCD services; (2) initiatives to recruit, train, motivate and supervise PHC workers with access to simplified guidelines; (3) access to essential medicines and technologies; (4) incorporation of NCD surveillance into health information systems and availability of region-specific monitoring mechanism; (5) creating information systems that enable reporting of disaggregated NCD data by equity domains (such as; age, sex, and geography) and (6) expanding UHC initiatives to reduce OOP expenditure. Combating NCDs is a political choice which need a whole-of-government approach.

## Author Contributions

**Conceptualization:** Azeb Gebresilassie Tesema, Seye Abimbola, Rohina Joshi, David Peiris.

**Data curation:** Azeb Gebresilassie Tesema, Whenayon Simeon Ajisegiri, Christine Balane.

**Formal analysis:** Azeb Gebresilassie Tesema, Whenayon Simeon Ajisegiri.

**Methodology:** Azeb Gebresilassie Tesema.

**Project administration:** Azeb Gebresilassie Tesema.

**Software:** Azeb Gebresilassie Tesema.

**Supervision:** Seye Abimbola, Padmanesan Narasimhan, Rohina Joshi, David Peiris.

**Validation:** Andre Pascal Kengne, Fassil Shiferaw, Jean-Marie Dangou, Rohina Joshi, David Peiris.

**Writing – original draft:** Azeb Gebresilassie Tesema.

**Writing – review & editing:** Azeb Gebresilassie Tesema, Whenayon Simeon Ajisegiri, Seye Abimbola, Andre Pascal Kengne, Fassil Shiferaw, Jean-Marie Dangou, Padmanesan Narasimhan, Rohina Joshi, David Peiris.

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
