## [Decision Letter · Decision Letter 0]

29 Jul 2020

PONE-D-20-19186

How well are non-communicable disease services being integrated into primary health care in Africa: a review of progress against World Health Organization’s African regional targets

PLOS ONE

Dear Dr. Tesema,

Thank you for submitting your manuscript to PLOS ONE. After careful consideration, we feel that it has merit but does not fully meet PLOS ONE’s publication criteria as it currently stands. Therefore, we invite you to submit a revised version of the manuscript that addresses the points raised during the review process.

The authors present a timely assessment of the progress being made in integrating NCD services into primary health care in Africa. Their review highlights the key deficiencies in this integration as very few countries have met the approved targets. The variable progress across this region highlights the considerable variability found across different countries in advancing progress in this domain.

Please find my comments and the comments of the other reviewers below.   Please pay particular attention to the comments which are listed as major and provide a detailed response describing how you intend to address those issues.

We look forward to receiving your revised manuscript.

Kind regards,

Sonak D. Pastakia

Academic Editor

PLOS ONE

Additional Editor Comments:

Methodology

Their combined research strategy of incorporating the feedback of the WHO Africa regional office is novel as it can help to validate and refine their findings with more in depth assessments from country based experts. There is a risk for bias but it seems this aspect of their methodology helped create a more comprehensive review of a difficult topic. At the same time, I do worry that some of the assessments might be biased as they are relying almost entirely on government reports of progress which would have an interest in presenting this data in a very positive light.  For example, I'm not sure that your approach for reporting availability of certain services such as medications is rigorous enough as it seems to largely be based on government-based reports rather than direct assessments of availability. Is there any way to assure the reader of the accuracy of your information and or the sources? At a minimum, this should be discussed as a limitation. (major)

One other thing which isn't entirely clear to me is how how your analysis goes beyond the WHO tracking efforts. What does the publication of this paper add to what is already presented by the WHO? It would be worthwhile to call that out specifically. (major)

If possible, I would have liked to see correlations with progress and NCD outcomes over time to potentially give positive feedback to countries that have made progress and suggestions for improvement to those who haven't. (minor)

Please also see the comments of the reviewers, especially reviewer 2, as addressing those comments will help to make this a stronger and more clear paper.

Journal Requirements:

2. Please upload a copies of Figures 2 and 3, to which you refer in your text on page 8. If the figures are no longer to be included as part of the submission please remove all reference to it within the text.

Reviewers' comments:

Reviewer's Responses to Questions

**Comments to the Author**

1. Is the manuscript technically sound, and do the data support the conclusions?

Reviewer #1: Yes

Reviewer #2: Yes

2. Has the statistical analysis been performed appropriately and rigorously? 

Reviewer #1: Yes

Reviewer #2: Yes

3. Have the authors made all data underlying the findings in their manuscript fully available?

Reviewer #1: Yes

Reviewer #2: Yes

4. Is the manuscript presented in an intelligible fashion and written in standard English?

Reviewer #1: Yes

Reviewer #2: Yes

5. Review Comments to the Author

Reviewer #1: This is a well researched and well written timely article. I only have minimal comments to add to this work.

Please add a definition of what the measure of mortality is that is reported at the beginning of the results section

The authors accurately point out a key limitation is the inability to verify progress vs implementation plans. The last sentence of the limitations (last paragraph in the discussion) implies that they were able to somewhat but not quite clear? Suggest you re-word for clarity. In addition, would suggest given where the study concludes, perhaps a suggestion of next steps to advance research in this field would be helpful.

The manuscript introduction and discussion sections have some minor grammar and punctuation errors rectifiable via a thorough edit e.g the first sentence of paragraph 7 of the discussion (page 10) is unclear.

Reviewer #2: I thank Editor for the opportunity to review this manuscript. I also thoroughly enjoyed reading the manuscript and commend the authors for providing incredibly interesting information and timely analyses/results regarding progress to strategically address NCD burden in Africa. I have a few minor comments that I hope the authors can address to strengthen the manuscript.

1. Overall, I was a little confused regarding which main framework was used for the analysis. It seemed that it was the Framework for Integrating Essential NCD Services at the PHC level with 4 targets. However, in the methodology and results section, it seemed that these indicators were fit into some of the WHO Health Systems Strengthening Building Blocks. When it came to the Discussion section, the authors went back to discuss their findings in the context of the 4 targets suggested by the WHO African Region Committee, without mentioning . I wasn't clear why the 12 indicators need to be "fit" into the WHO Building Blocks Framework. In my opinion, the Framework for Integrating Essential NCD Services at the PHC levels with 4 targets and 12 indicators are sufficient. If the authors are inclined to use the 6 building blocks, I think further discussion should be provided. (major)

2. The authors stated: "A review of national documents from all 47 countries in the WHO African Region was conducted. The detail review procedure is listed in the below sub-sections." I think this can be misleading because from my reading, it seemed clear that the authors analyze WHO reports primarily, not the national documents from each of these 47 countries. (please clarify, major)

3. Table 1: a few of the links led me to a website that was unrelated to the topic discussed (i.e. NCD Country Profiles) or led me to a website that could not be opened. Please double check your URLs. (major)

4. Table 2 (heat map): data were all presented as 0-3 (no data - fully achievement for the indicator). However, some of your indicators' answered are Yes/No questions (has a country trained or have plans to train PHC workers) or questions or questions answered in discrete values (# of essential medicines available out of 10). How did you translate those answers into 0-3 (major)

6. PLOS authors have the option to publish the peer review history of their article (what does this mean?). If published, this will include your full peer review and any attached files.

Reviewer #1: No

Reviewer #2: No

---

## [Author Response · Author response to Decision Letter 0]

30 Aug 2020

Author Response Letter: PLOS ONE [PONE-D-20-19186]

Dear Dr Pastakia, 

Thank you for reviewing our manuscript on “How well are non-communicable disease services being integrated into primary health care in Africa: a review of progress against World Health Organization’s African regional targets”. We have revised the manuscript to respond to the points raised by you and the reviewers and uploaded a clean and tracked change manuscript. 

Journal requirements

The manuscript is now formatted accordingly. 

• Please upload copies of Figures 2 and 3, to which you refer in your text on page 8. If the figures are no longer to be included as part of the submission, please remove all reference to it within the text.

Apologies - this referred to tables 2 and 3. We have replaced the word ‘figure’ with ‘table’ in the revised version, please see page 15; line 27. 

Academic Editor Comments

I. Methodology

Their combined research strategy of incorporating the feedback of the WHO Africa regional office is novel as it can help to validate and refine their findings with more in depth assessments from country-based experts. There is a risk for bias, but it seems this aspect of their methodology helped create a more comprehensive review of a difficult topic. At the same time, I do worry that some of the assessments might be biased as they are relying almost entirely on government reports of progress which would have an interest in presenting this data in a very positive light. For example, I'm not sure that your approach for reporting availability of certain services such as medications is rigorous enough as it seems to largely be based on government-based reports rather than direct assessments of availability. Is there any way to assure the reader of the accuracy of your information and or the sources? At a minimum, this should be discussed as a limitation. (major)

• We agree this review mainly relied on the WHO country’s progress reports and this may have introduced a bias. We have highlighted this as a potential limitation resulted from using these reports. We indicated this in the revised manuscript. Please see page 20; line 23-26. The implications of a positive reporting bias are that the large gaps we found could potentially be larger. As stated in the methods, we did also seek to interpret the findings drawing on empirical studies to provide greater depth to the country level findings (for example, indicator IV). However, this also comes with limitations – although internal validity may be enhanced, there is potential that external validity is compromised if the regions and populations studied are not representative at a national level.

II. One other thing which isn't entirely clear to me is how your analysis goes beyond the WHO tracking efforts. What does the publication of this paper add to what is already presented by the WHO? It would be worthwhile to call that out specifically. (major)

We believe this review adds value in three ways. First, the WHO NCDs reports are less granular, tracking all 9 NCDs voluntary global targets across multiple WHO regions to assess the overall national NCDs plan implementation progress, where health system strengthening is one of the targets. Our study had a more focussed scope reviewing African countries’ primary health care NCD implementation progress. Second, we explored associations between progress in these indicators and national health system capabilities including income status. Third, we also assessed relationships between each indicator and country level factors such as income status and out-of-pocket expenditure and found only weak associations on some parameters, suggesting that health system drivers to make progress in NCD targets are more complex than simply the wealth of the nation. We have added text regarding this on page 18; 1st paragraph of the discussion section. 

III. If possible, I would have liked to see correlations with progress and NCD outcomes over time to potentially give positive feedback to countries that have made progress and suggestions for improvement to those who haven't. (minor)

The Academic editor has raised an important research gap to explore in future, looking at the correlations between country progress and NCD outcomes at different time points. Time series analyses that sought to assess impact of certain policies on NCD outcomes would help shed light on this, however causal inference will always be contested given the complex and multifaceted nature of these policies and the timing of their introduction is not always easy to ascertain. 

Reviewers' comments:

Reviewer 1: 

IV. Please add a definition of what the measure of mortality is that is reported at the beginning of the results section 

Mortality estimates were taken from the WHO 2018 country profiles report and the measure used in the document was a proportional mortality (percentage of total death, in all ages and of both sex) for the four main NCDs. Please see page 10, line 9-10. 

V. The authors accurately point out a key limitation is the inability to verify progress vs implementation plans. The last sentence of the limitations (last paragraph in the discussion) implies that they were able to somewhat but not quite clear? Suggest you re-word for clarity. In addition, would suggest given where the study concludes, perhaps a suggestion of next steps to advance research in this field would be helpful.

We have included three recommendations for next steps: (1) strengthen independent monitoring mechanisms to assess implementation of NCDs interventions; (2) expand evaluation research in the field taking into consideration regional complexity; and (3) develop systems that enable reporting of NCD targets that are disaggregated by region. Please see page 21 last paragraph of limitations section. 

VI. The manuscript introduction and discussion sections have some minor grammar and punctuation errors rectifiable via a thorough edit e.g the first sentence of paragraph 7 of the discussion (page 10) is unclear.

We have clarified the idea in the revised version. Please see paragraph 7 of the discussion. Grammar and punctuation errors have been rectified throughout the paper. 

Reviewer 2: 

VII. Overall, I was a little confused regarding which main framework was used for the analysis. It seemed that it was the Framework for Integrating Essential NCD Services at the PHC level with 4 targets. However, in the methodology and results section, it seemed that these indicators were fit into some of the WHO Health Systems Strengthening Building Blocks. When it came to the Discussion section, the authors went back to discuss their findings in the context of the 4 targets suggested by the WHO African Region Committee, without mentioning. I wasn't clear why the 12 indicators need to be "fit" into the WHO Building Blocks Framework. In my opinion, the Framework for Integrating Essential NCD Services at the PHC levels with 4 targets and 12 indicators are sufficient. If the authors are inclined to use the 6 building blocks, I think further discussion should be provided. (major)

The main framework used was the WHO AFRO framework for Integration Essential NCDs Services at the PHC level, with its 4 targets which include PHC guidelines, workforce, medicines, and information systems. Indicators 1 – 10 fit well with 4 of the 6 WHO health systems building blocks, however, indicators 11 (out of pocket expenditure) and 12 (existence of an operational department) focus on the other two health system building blocks (financing and governance) and these are not present in the WHO AFRO framework. Given the aim of our study was to explore how NCDs are being integrated in the system we considered these two additional indicators were important to support a more holistic assessment. We agree it is a little confusing having two frameworks and so we have removed reference to the health system building blocks throughout the paper to make it clear and only referred to the WHO AFRO framework with the 4 targets and 12 indicators. 

VIII. The authors stated: "A review of national documents from all 47 countries in the WHO African Region was conducted. The detail review procedure is listed in the below sub-sections." I think this can be misleading because from my reading, it seemed clear that the authors analyse WHO reports primarily, not the national documents from each of these 47 countries. (please clarify, major)

We agree this is potentially misleading and have clarified this throughout the paper to indicate that the primary data source is the WHO reports. Please also see response 2 above to the academic editor’s query related to this.

IX. Table 1: a few of the links led me to a website that was unrelated to the topic discussed (i.e. NCD Country Profiles) or led me to a website that could not be opened. Please double check your URLs. (major)

Some of the data were extracted from the 2018 WHO NCDs country profile. However, in the revised manuscript, all the links/URLs have been checked. Please see the updated Table 1. 

X. Table 2 (heat map): data were all presented as 0-3 (no data - fully achievement for the indicator). However, some of your indicators answered are Yes/No questions (has a country trained or have plans to train PHC workers) or questions answered in discrete values (# of essential medicines available out of 10). How did you translate those answers into 0-3 (major)?

For most of the indicators, we used the 3-point scale to derive the heat map with a zero-value assigned if data were not available. For dichotomous value indicators, we gave a value of ‘3’ for ‘full achievement’, ‘1’ for ‘no achievement’ and ‘0’ for indicators with ‘no available data’ (see table below). Furthermore, we also present the actual numeric findings for each indicator in table 3. In the revised manuscript, we have edited Table 1 to make the measurement assumptions clearer. Please also see the method section in page 8. 

Indicators (variables that measures countries’ progress) Explanation 

For indicator I, II, VIII, IX, and indicator X 3= fully achieved

2= partially achieved 

1= not achieved 

0= No data found/not known

Indicator III 3= country has trained or plans to train PHC workers (Yes)

1= Data unavailable/country doesn’t have a strategy at all, or documents don’t indicate the PHC health worker training (this include document written in other languages) (No)

(The limitation of using this indicator is stated in page 12, line 25-28)

Indicator IV 3= national essential medicine is available, and available at PHCs (Yes at PHC)

2= national essential medicine/technology list is available (yes)

1 = if the country didn’t have an essential medicine/technology list or no list could be identified

Indicator V 3= All the 10 NCDs medicines were available 

2= 5-9 NCDs medicines were available

1= < 5 NCDs medicines were available 

0= No data available 

Indicator VI 3= All the 6 technologies were available 

2= 3-5 technologies were available 

1= <3 technologies were available 

0= No data available 

Indicator VII 3= countries with > 50% facilities provide service

2= countries with < 25 % facilities provide service

1= None/service not available

0= No data available 

Indicator XI 3= Less- if OOP < 20% 

2= Medium- if the OOP = 21-37%

1= Above the average (> 38% OOP)—if OOP is > 38%

0 = No data available/not know

Indicator XII 3= The country has a unit or other equivalent body in the country

1= No body available

Also please note that since we submitted this manuscript for publication, the latest 2020 WHO NCD progress report has been released and we have updated the data from the 2017 report for indicators (indicator I, VIII, IX and indicator X). The changes in the texts and tables are made accordingly (please also see table 2 and 3). There has been little change between 2017 and 2020 WHO country reports and the main findings including the interpretation of this study remain unchanged. 

Thank you again for your feedback and your time to review our paper. Please contact me if you have any further questions.

Best regards,

Azeb Gebresilassie Tesema 

ORCID iD: https://orcid.org/0000-0003-0618-4499

---

## [Decision Letter · Decision Letter 1]

30 Sep 2020

PONE-D-20-19186R1

How well are non-communicable disease services being integrated into primary health care in Africa: a review of progress against World Health Organization’s African regional targets

PLOS ONE

Dear Dr. Tesema,

Thank you for submitting your manuscript to PLOS ONE. After careful consideration, we feel that it has merit but does not fully meet PLOS ONE’s publication criteria as it currently stands. Therefore, we invite you to submit a revised version of the manuscript that addresses the points raised during the review process.

Thank you for your revisions.  Please see the few remaining comments as the paper is nearly ready for acceptance once those minor comments are addressed.

We look forward to receiving your revised manuscript.

Kind regards,

Sonak D. Pastakia

Academic Editor

PLOS ONE

Additional Editor Comments (if provided):

Thank you for your responses to the suggested revisions. The paper is considerably improved and nearly ready for publication once the few minor comments are resolved.

Reviewers' comments:

Reviewer's Responses to Questions

**Comments to the Author**

1. If the authors have adequately addressed your comments raised in a previous round of review and you feel that this manuscript is now acceptable for publication, you may indicate that here to bypass the “Comments to the Author” section, enter your conflict of interest statement in the “Confidential to Editor” section, and submit your "Accept" recommendation.

Reviewer #1: All comments have been addressed

Reviewer #2: All comments have been addressed

2. Is the manuscript technically sound, and do the data support the conclusions?

Reviewer #1: Yes

Reviewer #2: Yes

3. Has the statistical analysis been performed appropriately and rigorously? 

Reviewer #1: Yes

Reviewer #2: Yes

4. Have the authors made all data underlying the findings in their manuscript fully available?

Reviewer #1: Yes

Reviewer #2: Yes

5. Is the manuscript presented in an intelligible fashion and written in standard English?

Reviewer #1: Yes

Reviewer #2: Yes

6. Review Comments to the Author

Reviewer #1: (No Response)

Reviewer #2: Thank you very much for taking the time to respond to my comments/questions. I believe the revised manuscript provided much needed clarity on areas of prior concerns, including data sources, collection, and analysis. I have very few minor comments that should be addressed prior to publication:

1. Please provide the full name of WHO PEN (WHO Package of Essential Noncommunicable disease interventions), in addition to the abbreviation, the first time you mentioned it in the manuscript (introduction).

2. Table 1 - Target 3 - URL for "Data was extracted from WHO essential medicine and health product information portal" - The page cannot be found. Please double check your URL.

3. Table 3 - Indicator "Availability of essential medicine list at national and PHC level," please change your subtext to reflect essential medicine LIST. For example, "Essential medicine LIST available nationally and at the PHC level."

Thank you!

7. PLOS authors have the option to publish the peer review history of their article (what does this mean?). If published, this will include your full peer review and any attached files.

Reviewer #1: No

Reviewer #2: No

---

## [Author Response · Author response to Decision Letter 1]

3 Oct 2020

Date: October 3, 2020

Author Response Letter: PLOS ONE [PONE-D-20-19186R1 and PONE-D-20-19186R2] 

Dear Dr Pastakia, 

Thank you for reviewing the revised manuscript on “How well are non-communicable disease services being integrated into primary health care in Africa: a review of progress against World Health Organization’s African regional targets”. We have made the revision based on the points the reviewer raised and uploaded a clean and tracked change manuscript. 

Reviewer' comments

1. Please provide the full name of WHO PEN (WHO Package of Essential Noncommunicable disease interventions), in addition to the abbreviation, the first time you mentioned it in the manuscript(introduction).

• We have now included the full name of WHO PEN in the revised manuscript. Please see page 3, line 22. 

2. Table 1 - Target 3 - URL for "Data was extracted from WHO essential medicine and health product information portal" - The page cannot be found. Please double check your URL.

• The correct URL has included now, where it shows the National (all countries) medicines list/formulation/standard treatment guideline portal. Please see Table 1, Target 3. 

3. Table 3 - Indicator "Availability of essential medicine list at national and PHC level," please change your subtext to reflect essential medicine LIST. For example, "Essential medicine LIST available nationally and at the PHC level."

• We have made the correction in the revised manuscript. Please see Table 3. 

Editor’s comment [PONE-D-20-19186R2]

1. Your ethics statement should only appear in the Methods section of your manuscript. If your ethics statement is written in any section besides the Methods, please move it to the Methods section and delete it from any other section. Please ensure that your ethics statement is included in your manuscript, as the ethics statement entered into the online submission form will not be published alongside your manuscript.

• We have moved the ethics statement to the method section of the manuscript. Please see page 11, line 24. 

Thank you again for your feedback and your time to review our paper. Please contact me if you have any further questions.

Best regards,

Azeb Gebresilassie Tesema 

ORCID iD: https://orcid.org/0000-0003-0618-4499

---

## [Editor Report · Decision Letter 2]

7 Oct 2020

How well are non-communicable disease services being integrated into primary health care in Africa: a review of progress against World Health Organization’s African regional targets

PONE-D-20-19186R2

Dear Dr. Tesema,

We’re pleased to inform you that your manuscript has been judged scientifically suitable for publication and will be formally accepted for publication once it meets all outstanding technical requirements.

Kind regards,

Sonak D. Pastakia

Academic Editor

PLOS ONE

Additional Editor Comments (optional):

Thank you for making the few remaining changes.  This final version is much improved from the original and a valuable contribution to the literature.
---

## [Editor Report · Acceptance letter]

13 Oct 2020

PONE-D-20-19186R2 

How well are non-communicable disease services being integrated into primary health care in Africa: a review of progress against World Health Organization’s African regional targets 

Dear Dr. Tesema:

I'm pleased to inform you that your manuscript has been deemed suitable for publication in PLOS ONE. Congratulations! Your manuscript is now with our production department. 

Kind regards, 

on behalf of

Dr. Sonak D. Pastakia 

Academic Editor

PLOS ONE